# Tea Bag Index to Assess Carbon Decomposition Rate in Cranberry Agroecosystems

**Wilfried Dossou-Yovo [1,2,*], Serge-Étienne Parent [1], Noura Ziadi [2], Élizabeth Parent [1] and Léon-Étienne Parent [1]**

[1] Department of Soils and Agri-Food Engineering, Université Laval, Quebec, QC G1V 0A6, Canada; serge-etienne.parent@usherbrooke.ca (S.-É.P.); Elizabeth.Parent@fsaa.ulaval.ca (É.P.); Leon-Etienne.Parent@fsaa.ulaval.ca (L.-É.P.)

[2] Agriculture and Agri-Food Canada, Quebec Research and Development Center, 2560 Hochelaga Boulevard, Quebec, QC G1V 2J3, Canada; noura.ziadi@canada.ca

[*] Correspondence: wilfrieddossouyovo16@gmail.com

**Abstract:** In cranberry production systems, stands are covered by 1–5 cm of sand every 2–5 years to stimulate plant growth, resulting in alternate layers of sand and litter in soil upper layers. However, almost intact twigs and leaves remain in subsurface layers, indicating a slow decomposition rate. The Tea Bag Index (TBI) provides an internationally standardized methodology to compare litter decomposition rates (k) and stabilization (S) among terrestrial ecosystems. However, TBI parameters may be altered by time-dependent changes in the contact between litter and their immediate environment. The aims of this study were to determine the TBI of cranberry agroecosystems and compare it to the TBI of other terrestrial ecosystems. Litters were standardized green tea, standardized rooibos tea, and cranberry residues collected on the plantation floor. Litter decomposition was monitored during two consecutive years. Added N did not affect TBI parameters (k and S) due to possible N leaching and strong acidic soil condition. Decomposition rates (k) averaged (mean $\pm$ SD) $9.7 \times 10^{-3}$ day$^{-1}$ $\pm 1.6 \times 10^{-3}$ for green tea, $3.3 \times 10^{-3}$ day$^{-1}$ $\pm 0.8 \times 10^{-5}$ for rooibos tea, and $0.4 \times 10^{-3}$ day$^{-1}$ $\pm 0.86 \times 10^{-3}$ for cranberry residues due to large differences in biochemical composition and tissue structure. The TBI decomposition rate (k) was 0.006 day$^{-1}$ $\pm 0.002$ in the low range among terrestrial ecosystems, and the stabilization factor (S) was $0.28 \pm 0.08$, indicating high potential for carbon accumulation in cranberry agroecosystems. Decomposition rates of tea litters were reduced by fractal coefficients of 0.6 for green tea and 0.4 for rooibos tea, indicating protection mechanisms building up with time in the tea bags. While the computation of the TBI stabilization factor may be biased because the green tea was not fully decomposed, fractal kinetics could be used as additional index to compare agroecosystems.

**Keywords:** carbon flux; fractal kinetics; nitrogen fertilization; Tea Bag Index; podzols; gleysols

## 1. Introduction

Soil is the largest pool of terrestrial organic carbon (C) in the biosphere, storing more carbon than plants and the atmosphere combined [1]. Increasing soil organic carbon content plays a key role in enhancing soil quality and mitigating greenhouse gases [2–4]. Soil carbon storage capacity depends on temperature, rainfall, decomposing organisms, substrate quality [5–10], and cultural practices [11–15].

Wisconsin, USA; Quebec, Canada; and Massachusetts, USA, are the world leaders in cranberry production [16]. Most cranberry stands are established on constructed anthropogenic acidic sandy soils [17,18]. Cranberry is a low growing, trailing, broadleaf and leathery, woody, non-deciduous vine [19]. Stolons or runners range from one to two meters long and the uprights originating from stolons are 5–20 cm long. Stolons and uprights form a thick and tough mat covering the whole surface of the field. Nitrogen, applied at rates that may vary in the range of 15 to 60 kg N ha$^{-1}$, is the nutrient impacting the

most vegetative growth, sometimes reaching excessive biomass production [20]. Cranberry beds are covered every two to five years by 1–5 cm of sand to stimulate or maintain crop productivity [19], resulting in alternate thin sublayers of sand and organic matter in the root zone [18,21,22]. Cranberry twigs and leaf residues appear almost intact at depth [21], indicating a slow decomposition rate. Similar to natural depositional processes [23], burying activities can, thus, contribute to soil carbon storage in cranberry agroecosystems.

Biochemical quality of the litter is the main driver of litter decomposition under comparable climate conditions [24–26]. The carbon sink from ligneous plants is large per unit of nitrogen deposited [12,27]. The C:N ratio is the most frequent index of decomposability of organic substrates [28]. While nitrogen is added in amounts generally ranging from 30 to 50 kg N ha$^{-1}$ in cranberry production systems [29,30] to reach tissue nitrogen levels of 0.90–1.10% [31], excessive nitrogen fertilization may lead to vegetative overgrowth [30], possibly impacting litter quality, C:N ratio, decomposition rate, and carbon accumulation. The C:N ratio of fruiting and vegetative uprights varies between 50 and 80 at mid-fruit development depending on N fertilization [32].

Litterbags [33–35] and cotton strips [36–39] have been used to monitor litter decomposition and assess carbon decomposition rates [40–43]. While cotton strips are less representative of litter biochemical compositions [39,43], bags of native plant litter and standardized substrates allow for comparing litter decomposition rates at regional and global scales [44,45]. The Tea Bag Index (TBI) has been designed to collect litter decomposition data uniformly across ecosystems. The TBI is a standard measure of mass loss of green *(Camellia sinensis)* and rooibos *(Aspalathus linearis)* teas after 90 days of incubation in the soil [44]. The green tea, a labile litter, and the rooibos, a more recalcitrant litter, show contrasting decomposition rates. The TBI is determined after 90 days of incubation from decomposition rate (k) by mass loss and a stabilization factor (S) depending on environmental conditions during the conversion of labile to recalcitrant compounds.

Organic matter decomposition is generally assessed by first-order kinetics assuming that the reaction rate is constant through time [46]. However, the reactive surfaces of organic matter may become less accessible to enzymes and soil microorganisms as protection mechanisms build up during decomposition, leading to departure from first-order kinetics, a process known as fractal-like kinetics [2,47,48]. Fractal kinetics has been used to describe the complex kinetic of enzymatic saccharification of cellulose, and the elimination of lignin [49,50]. Fractal kinetics of green and rooibos decomposition may differ among ecosystems, possibly impacting their TBI classification.

We hypothesized that (1) the decomposition rate of cranberry residues is low compared to decompositions rates of green and rooibos teas, (2) the TBI of the cranberry agroecosystems is stable across years, (3) the TBI of the cranberry agroecosystems is low compared to those of other ecosystems whatever the nitrogen fertilization regime, and (4) decomposition rates of tea litters follow fractal-like kinetics. The aim of this study was to assess the carbon decomposition rate in cranberry agroecosystems compared to other terrestrial ecosystems.

## 2. Materials and Methods

### 2.1. Study Area

This study was conducted at four sites during growing seasons 2017 and 2018 in Southern Quebec, Canada (Figure 1), providing eight site × year combinations. There were three conventionally managed cranberry sites, site #45 (46°16′34.7′′ N; 71°51′30.0′′ W), site #9 (46°16′39.4′′; 71°52′14.2′′ W), site #10 (46°19′28.7′′ N; 71°44′41.6′′ W), and one organically managed cranberry site, site #A9 (46°14′16.5′′ N; 72°02′13.4′′ W). Sites #9, #45, #A9, and #10 have been planted with the "Stevens" cultivar in 1995, 1999, 2004, and 2007, respectively. At the establishment phase, the soils had been capped by 60–75 cm of fine to coarse sand over the sandy floor after removing the upper layers (Horizons A and B, and part of the C horizon). Soil series on sites #A9, #45, and #9 were Saint-Jude and Sainte-Sophie (Humo-Ferric Podzol in the Canadian System, Haplorthods in the U.S. Soil

Taxonomy, Orthic Podzol in the World Reference Base for Soil Resources), and Saint-Samuel (Humic Gleysol in the Canadian System; Humaquepts in the U.S. Soil Taxonomy; Umbric Gleysol in the World Reference Base for Soil Resources) on site #10 [51].

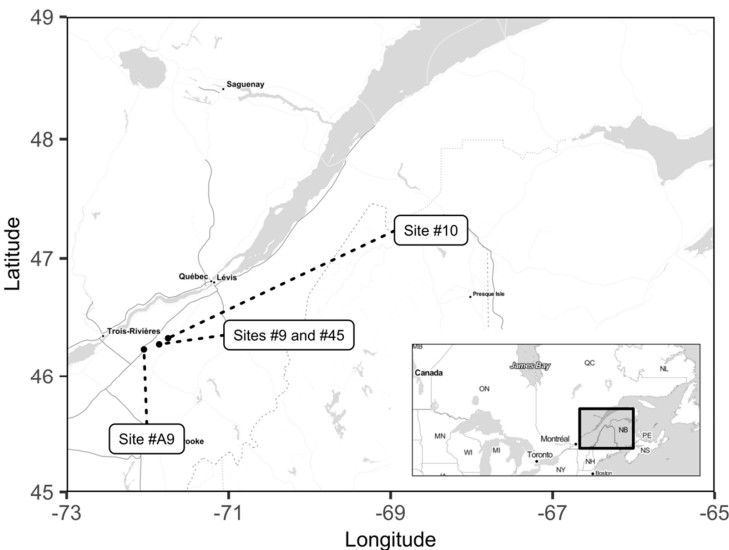

**Figure 1.** Map of the cranberry experimental sites in Southern Quebec, Canada.

The climate of the region is sub-humid temperate and continental with cold winters and hot summers (Figure 2). Year 2017 was drier than normal during July and August. Cranberry beds were sprinkler-irrigated to maintain the soil matric potential between −4 and −7 kPa [52].

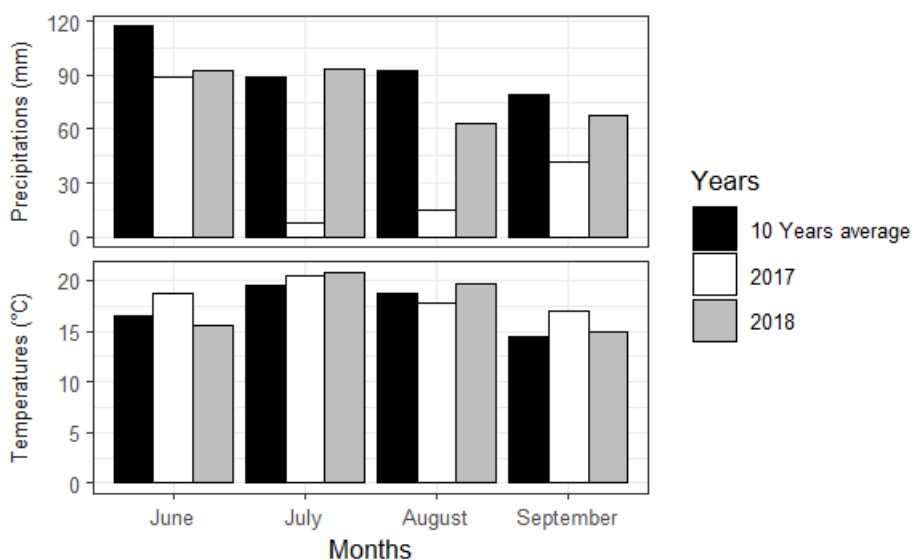

**Figure 2.** Growing-season months (2017–2018) and 10-year averages of weather data at Lemieux, Quebec (46°09′56″ N 72°19′28″ W), near experimental sites [53].

### 2.2. Soil Analysis

Soils were sampled in the 0–15 cm layer in spring 2017. Samples were air-dried and passed through a 2-mm sieve. Soil pH was measured in 0.01 M $CaCl_2$ (soil to solution ratio of 1:2 *v:v*). The total carbon, nitrogen, and sulfur were quantified by combustion [54] using the Leco CNS model 630-300-200 (Leco Corporation, Saint-Joseph, MI). Soil texture was measured by sedimentation [55] followed by hand-sieving sand particles. The soil's nutrient contents were extracted using the Mehlich-3 method [56] and quantified by inductively

coupled plasma—optical emission spectrometry (Perkin Elmer, Waltham, MA, USA). Soil properties are presented in Table 1.

**Table 1.** Soil chemical properties at experimental sites (0–15 cm layer).

| | Site #A9 | Site #10 | Site #9 | Site #45 |
|---|---|---|---|---|
| $pH_{cacl2}$ | 4.3 | 4.3 | 4.7 | 4.6 |
| C:N | 19.0 | 7.7 | 11.0 | 21.8 |
| | g kg$^{-1}$ | | | |
| C | 11.4 | 5.4 | 12.1 | 13.1 |
| N | 0.6 | 0.7 | 1.1 | 0.6 |
| S | 0.2 | 0.2 | 0.1 | 0.1 |
| Sand 1–2 mm | 13 | 6 | 34 | 19 |
| Sand 0.5–1 mm | 71 | 28 | 93 | 74 |
| Sand 0.25–0.5 mm | 411 | 329 | 404 | 279 |
| Sand 0.1–0.25 mm | 375 | 461 | 261 | 475 |
| Sand 0.25–0.01 mm | 44 | 102 | 69 | 91 |
| Silt | 49 | 5 | 44 | 37 |
| Clay | 36 | 23 | 26 | 26 |
| | Mehlich-3 extractable mg element kg$^{-1}$ | | | |
| P | 47 | 46 | 88 | 62 |
| S | 39 | 25 | 25 | 38 |
| K | 26 | 23 | 36 | 35 |
| Ca | 28 | 26 | 170 | 135 |
| Mg | 6 | 7 | 12 | 10 |
| Zn | 0.5 | 0.7 | 2.0 | 1.1 |
| Cu | 1.5 | 2.5 | 2.1 | 1.9 |
| Mn | 0.8 | 0.9 | 3.0 | 1.0 |
| Fe | 128 | 108 | 159 | 182 |
| Al | 591 | 345 | 690 | 662 |
| | Mehlich-3 P saturation ratio | | | |
| P/Al | 0.08 | 0.13 | 0.13 | 0.09 |

*2.3. Litter Bags*

Green tea (EAN 87 22700 05552 5) and rooibos tea (EAN 87 22700 18843 8) are two commercially available teas type (Lipton, Westervoort, The Netherlands) used to measure TBI (http://www.teatime4science.org/) (accessed on 4 August 2021). For comparison, we also used cranberry residues made of a mixture of vegetative and fruiting uprights collected on the plantation floor and cut to a particle size less than 5 mm. Hence, cranberry litter composition (stems and leaves from vegetative and fruiting uprights) varied between years. Particle-size distribution of the litters (Table 2) was determined by wet sieving [57]. At the onset of the experiments, the C:N ratios (mean $\pm$ SD) were 11.1 $\pm$ 1.3 and 54.01 $\pm$ 9.1 for green tea and rooibos tea, respectively. The C:N ratios for the teas differed slightly from those reported by Keuskamp et al. [44]. The C:N ratio of the cranberry residues was 66.7 $\pm$ 5.3 in the experiments, within the range of the C:N ratios reported by [32] for cranberry uprights. Cranberry residues were used for comparison only and were not considered in the computation of the TBI.

**Table 2.** Particle size distribution of the control tea litters and cranberry residues.

| Tea Type | Size Fraction (mm) | Particle Size Distribution (%) |
|---|---|---|
| Green tea | >2 | 32.1 |
| | 1–2 | 26.2 |
| | 0.5–1 | 5.9 |
| | 0.25–0.50 | 1.7 |
| | <0.25 | 34.1 |
| Rooibos tea | >2 | 64.8 |
| | 1–2 | 14.4 |
| | 0.5–1 | 1.3 |
| | 0.25–0.50 | 0.5 |
| | <0.25 | 19.0 |
| Cranberry residue | >2 | 89.2 |
| | 1–2 | 0.1 |
| | 0.5–1 | - |
| | 0.25–0.50 | - |
| | <0.25 | 10.7 |

Litters were oven-dried at 65 °C for 48, and then ground in a coffee mill to <2 mm for biochemical analysis, using the Ankom$^{200/220}$ fiber analyzer [58,59]. The biochemical fractions analyzed were the soluble, hemicellulose, cellulose, and lignin+cutin fractions. The holocellulose fraction is the sum of the hemicellulose and cellulose fractions. The hydrolyzable fraction is the sum of the soluble and holocellulose fractions.

In total, 3792 bags were buried at a depth of 8 cm in cranberry beds in 2017 and 2018. The number of bags was much larger than the 110 to 234 bags used for within one-year ecological [60,61] and laboratory [62] experiments due to the bags being damaged during unearthing or not being found in the tough cranberry mat, the high requirement for model testing, and the need to composite samples to run biochemical analyses.

*2.4. Experimental Designs*

The trials were conducted during two consecutive years to test the constancy of the TBI parameters through time. Because fertilization regimes may vary among cranberry production sites depending on the cultivar and soil conditions and impact the C:N ratio of cranberry residues [32], in the first year, we also tested whether nitrogen regimes established in 2014 could influence the TBI. The nitrogen was applied at rates of 30, 45, and 60 kg N ha$^{-1}$ within the range for cranberry [32]. Nitrogen sources were ammonium sulfate in the conventional sites or fish emulsions (6-1-1) in the organic site. Nitrogen was applied at four occasions during the season: 15% at early flowering (29 June to 2 July), 35% at 50% flowering (8 to 11 July), 35% at 50% fruit set (16 to 19 July), and 15% at 1 to 2 weeks after the third application. Other nutrients were applied according to local fertilization trials or regional fertilizer recommendations. In the second year, we tested whether litter decomposition rate could be fractal-like because green tea was not fully decomposed, as suggested by Keuskamp et al. [44], and compacted through time, indicating reduced contact between substrates and their immediate environment.

2.4.1. Year 1 Experiment to Test TBI Stability across Fertilization Regimes

Samples of 1.7–1.9 g of litter were introduced into tetrahedral polyethylene tea bags (5 cm × 5 cm; 0.25 mm mesh size) and sealed with an impulse heat sealer. Each bag and litter were weighed and tagged. The setup was a complete random design with three (03) nitrogen fertilization doses replicated four (04) times. There were three (03) litter types (green and rooibos teas and cranberry residues) replicated three (03) times in each plot. There was a total of 108 bags per site, and 432 bags across the four (04) sites. Plot size was 12 m$^2$ (3 m × 4 m). Bags were buried at a depth of 8 cm on 15 June 2017, and collected 90 days later, as prescribed by Keuskamp et al. [44], on 15 September 2017. Cranberry

stands form tough terrain. We collected 271 bags because 161 bags were not found in the ground or were torn during unearthing. The tea bags were dried at 65 °C for 48 h. Adhering soil particles were gently removed from the bag surfaces with a brush. Litter was weighed, ground in a coffee mill to <2 mm, and analyzed for total carbon by Dumas combustion (Kowalenko, 2001) to measure carbon loss during the season.

### 2.4.2. Year 2 Experiment to Validate TBI across Years and Test Fractal Behavior

Bags of tea and cranberry residues were buried from 14 May to 8 October 2018, at the four sites. The design was a combination of four sites, three litter types (green and rooibos teas and cranberry residues) repeated 40 times and collected at 21, 42, 63, 84, 105, 126, and 147 days (3360 bags in total) at each of the four sites. We collected 2613 bags because 747 bags were not found in the ground or were torn upon unearthing. After bag collection, litter samples were dried at 65 °C, and ground in a coffee mill to pass through 2-mm sieve. The ground litter was analyzed for total carbon by Dumas combustion [54]. Bags were composited for each combination of litter type and period to provide enough material to perform biochemical analyses by Fourier-transformed near-infrared spectroscopy (Nicolet Antaris FT-NIR analyzer, Thermo Electron Corp., Ann Arbor, MI, USA) at each site [63].

### 2.4.3. Reaction Kinetics

Decomposition rates of litter materials (cranberry residues, green and rooibos tea) were estimated from two points (initial and final mass), as follows:

$$\ln(M_t/M_0) = -kt$$

where $M_0$ is the initial litter biomass, $M_t$ is the litter biomass that remained after incubation time t = 90 days, and k is the decomposition rate.

Depending on the nature of the materials being processed and the immediate environment, the reaction rate (k) may decrease through time, as follows [48]:

$$k_t = k_1 \, t^{-h} \tag{1}$$

Hence,

$$\log(k_t) = -h \log(t) + \log(k_1) \tag{2}$$

The first order kinetic becomes fractal-like, as follows:

$$\ln(M_t/M_0) = - \, k_1 \, t^{\,1-h} \tag{3}$$

where $k_1$ is the reaction rate at time t = 1 and h is the fractal coefficient ($0 \le h \le 1$, $t \ge 1$) that accounts for the reaction rate decelerating with time.

### 2.4.4. TBI

The TBI assumes that the decomposition rate follows an exponential decay from the remaining rooibos biomass after 90 days of incubation. The TBI was computed as follows (Keuskamp et al., 2013):

$$M_t = a_r \, e^{-k_1 \, t} + (1 - a_r) \times (e^{-k_2 \, t}) \tag{4}$$

where $M_t$ is the fraction of the rooibos biomass that remains after incubation time t, $a_r$ is the decomposable fraction of the rooibos tea estimated from its hydrolyzable fraction and the stabilization factor assessed for green tea decomposition, $k_1$ is the decomposition rate of the labile fraction, and $k_2$ is the decomposition rate of the recalcitrant fraction, considered to be very low compared to $k_1$, hence neglected,

$$k_1 = \ln(a_r/M_t - (1 - a_r))/t \tag{5}$$

The stabilization factor (S) was computed as follows:

$$S = 1 - a_g/H_g \tag{6}$$

where $a_g$ and $H_g$ are the decomposable (after 90 days) and the van Soest hydrolyzable fractions of green tea, respectively. The decomposable fraction of rooibos tea ($a_r$) was predicted from the van Soest hydrolyzable fraction of rooibos tea ($H_r$) and S as follows:

$$a_r = H_r \times (1 - S) \tag{7}$$

### 2.4.5. Statistical Analysis

Statistical analyses were performed in the R environment 3.6.2 version [64]. The lme function of the nlme package 3.1.143 version [65] provides reliable tools to run the model [66]. Compatibility intervals [67] were computed at the 0.05 probability level.

## 3. Results

### 3.1. Organic Matter Decomposition

The biochemical fractions of litters are presented in Table 3. Rooibos and green teas contained similar amounts of the hydrolyzable fraction, which was much higher compared to cranberry litter. Green tea decomposed fastest due to the high proportion of soluble materials. Rooibos tea contained the largest proportion of holocellulose and showed intermediate decomposition rates. Cranberry litter showed the highest lignin+cutin content, and the slowest decomposition rate. Decomposition rates for rooibos tea and cranberry litter were, respectively, 0.34 and 0.04 times that of green tea. Mass loss averaged 59% for green tea, 26% for rooibos tea, and 5% for cranberry litter after 90 d of incubation.

**Table 3.** Properties of teas and litter materials (mean ± SD) in the first experiment performed in 2017.

| | | Green Tea | Rooibos Tea | Cranberry Litter |
|---|---|---|---|---|
| | C:N Ratio | 11.1 ± 1.3 | 54.1 ± 9.1 | 66.7 ± 5.4 |
| | | % of dry organic mass (total biomass minus ash) (mean ± SD) | | |
| 1. | Soluble matter | 75.2 ± 1.5 | 45.4 ± 1.4 | 33.9 ± 3.3 |
| 2. | Holocellulose | 14.8 ± 0.9 | 37.5 ± 0.3 | 25.9 ± 0.9 |
| 3. | Lignin-cutin | 17.1 ± 1.1 | 9.8 ± 1.4 | 40.1 ± 2.7 |
| | Hydrolyzable (1 + 2) | 90.5 ± 2.4 | 82.9 ± 1.7 | 47.6 ± 4.2 |
| | | Biomass weight (g) (mean ± SD) | | |
| | $M_0$ [†] | $1.7 \pm 1.3 \times 10^{-2}$ | $1.9 \pm 1.1 \times 10^{-2}$ | $1.9 \pm 1.5 \times 10^{-2}$ |
| | $M_{90 \text{ days}}$ | $0.7 \pm 1.2 \times 10^{-1}$ | $1.4 \pm 1.1 \times 10^{-1}$ | $1.8 \pm 1.3 \times 10^{-1}$ |
| | | Decomposition rate k (day$^{-1}$) as ($\ln(M_{90 \text{ days}}/M_0)$)/90 days (mean ± SD) | | |
| | | $9.7 \times 10^{-3} \pm 1.6 \times 10^{-3}$ | $3.3 \times 10^{-3} \pm 0.8 \times 10^{-3}$ | $0.4 \times 10^{-3} \pm 0.86 \times 10^{-3}$ |

† Initial $M_0$ and residual $M_{90 \text{ days}}$ masses of plant materials, respectively.

The nitrogen dosage showed negligible effects on the TBI decomposition rates ($p$-value = 0.149) and stabilization factor ($p$-value = 0.530) (Figures 3 and 4). Site #10 showed the lowest litter decomposition rate. Site #9 showed the lowest stabilization factor. As a result, differences in the TBI among sites are attributable to local factors other than N fertilization.

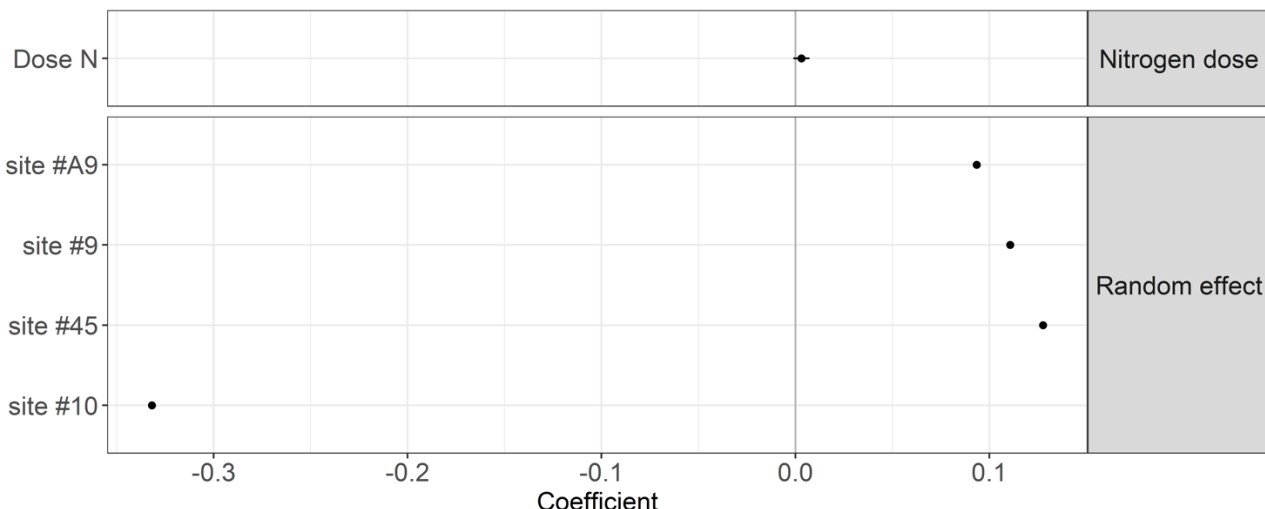

**Figure 3.** Coefficients of the mixed model linking the TBI decomposition rate (k) to the N dosage and sites with a 95% compatibility interval.

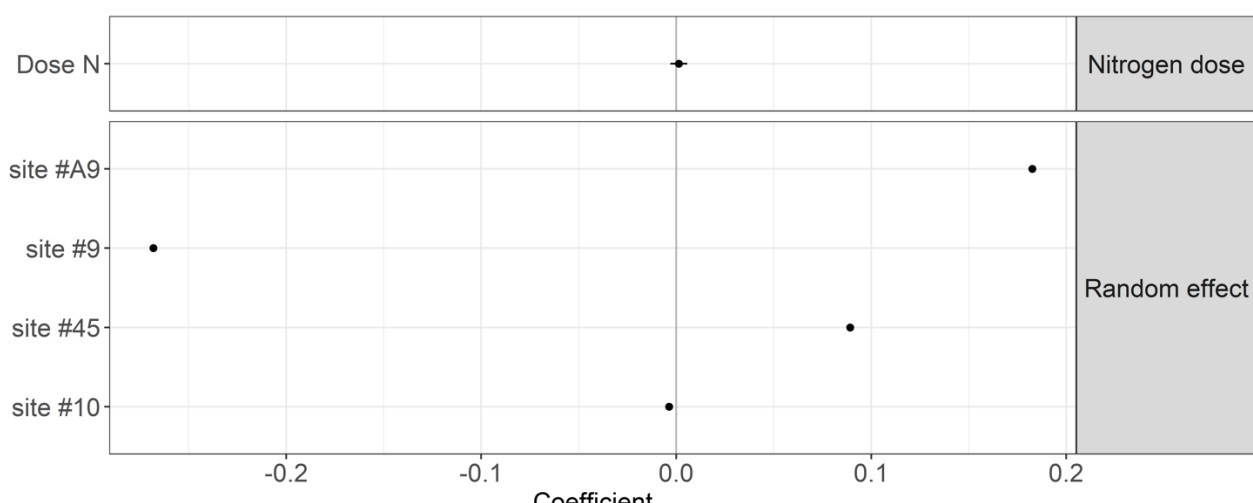

**Figure 4.** Coefficients of the mixed model linking the TBI stabilization factor (S) to the N dosage and sites with a 95% compatibility interval.

In 2018, two-point decomposition rates between 0 and 90 days of incubation were (mean $\pm$ SD) as follows: $10.5 \times 10^{-3}$ day$^{-1}$ $\pm$ $5.5 \times 10^{-3}$ for green tea, $3.9 \times 10^{-3}$ day$^{-1}$ $\pm$ $1.4 \times 10^{-3}$ for rooibos tea, and $0.8 \times 10^{-3}$ day$^{-1}$ $\pm$ $0.9 \times 10^{-3}$ for cranberry residues. The corresponding TBI parameters of the cranberry agroecosystems were high for S ($0.287 \pm 0.08$) and very low for k ($0.006$ day$^{-1}$ $\pm$ $0.002$) compared to those of ecosystems documented by Keuskamp et al. [44] (Figure 5).

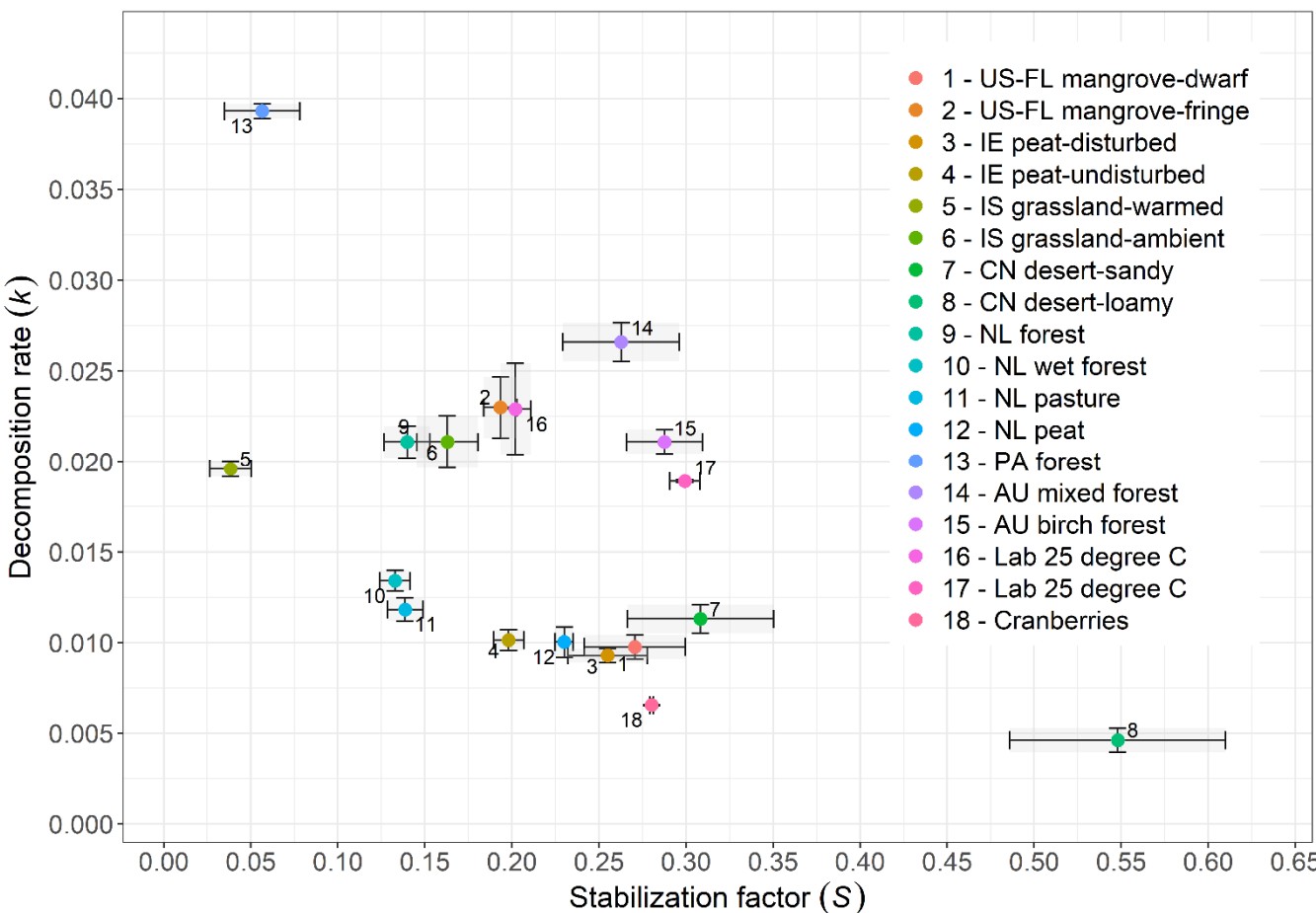

**Figure 5.** TBI of Quebec cranberry agroecosystems compared to the TBI of terrestrial ecosystems reported by Keuskamp et al. (2013). United States–Florida = US-FL, China = CN, Panama = PA, the Netherlands = NL, Austria = AU, Ireland = IE, and Iceland = IS.

### 3.2. Fractal Kinetics of Litter Decomposition

There was a small difference between litter decomposition rates recorded in 2017 (Table 3) and 2018 (Figure 6), indicating a consistent TBI across years. On the other hand, decomposition rates of green and rooibos teas were found to decrease non-linearly with time, indicating fractal-like kinetics (Figure 6). Decomposition rates of green and rooibos teas decelerated by factors of 2.7 and 1.6, respectively, at the end of 147 d of incubation, compared to the initial decomposition rate. The decomposition kinetics of green and rooibos teas were, thus, fractal-like, decreasing non-linearly with time, with fractal coefficients of 0.6 and 0.4, respectively (Figure 7). The decomposition rate of the cranberry litter tended to increase slightly after 63 days of incubation, indicating gradual opening of tissue internal structure by the microflora.

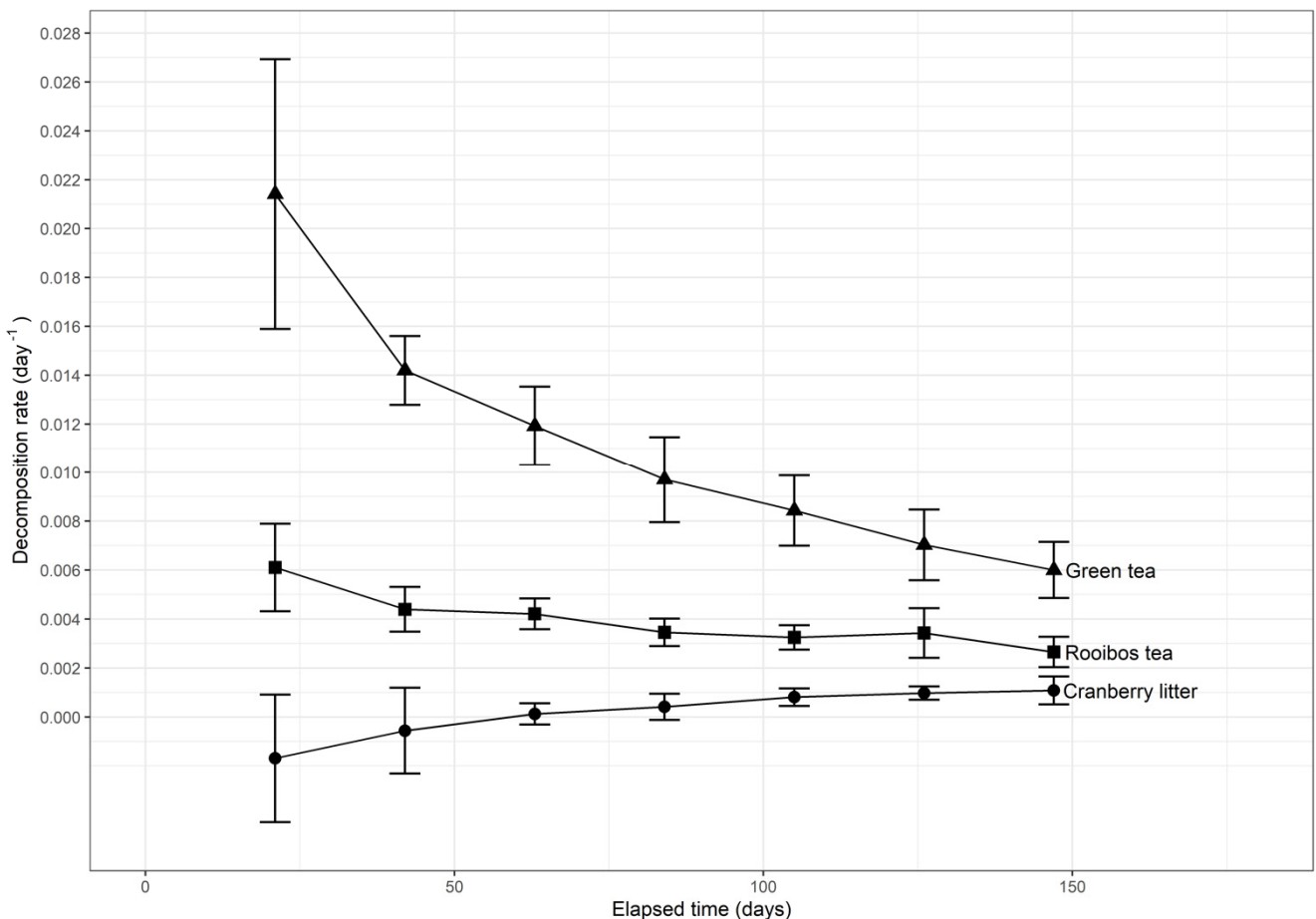

**Figure 6.** First order kinetics of teas and cranberry litter under field conditions.

The decomposition patterns (initial decomposition rate $k_1$ and fractal coefficient h) of the hydrolyzable (soluble matter and holocellulose) and non-hydrolyzable (lignin and cutin) fractions are presented in Figure 8. The hydrolyzable fraction of green tea showed higher fractal coefficients (h = 0.6) than that of rooibos tea (h = 0.4), with both different from zero. The fractal coefficient for the non-hydrolyzable fraction of green tea and rooibos tea was h = 1.2 and h = 1.7, respectively, but were not different from one, indicating considerable deceleration through time and great potential for carbon accumulation.

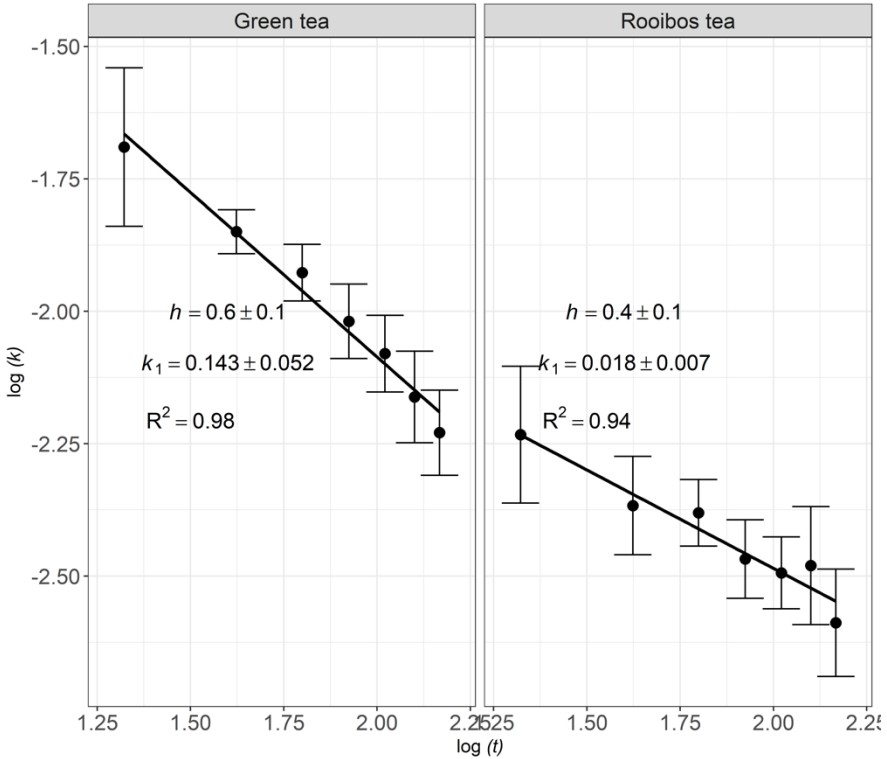

**Figure 7.** Fractal kinetics of the tea's decomposition under field conditions with a 95% compatibility interval for the experiment performed in 2018.

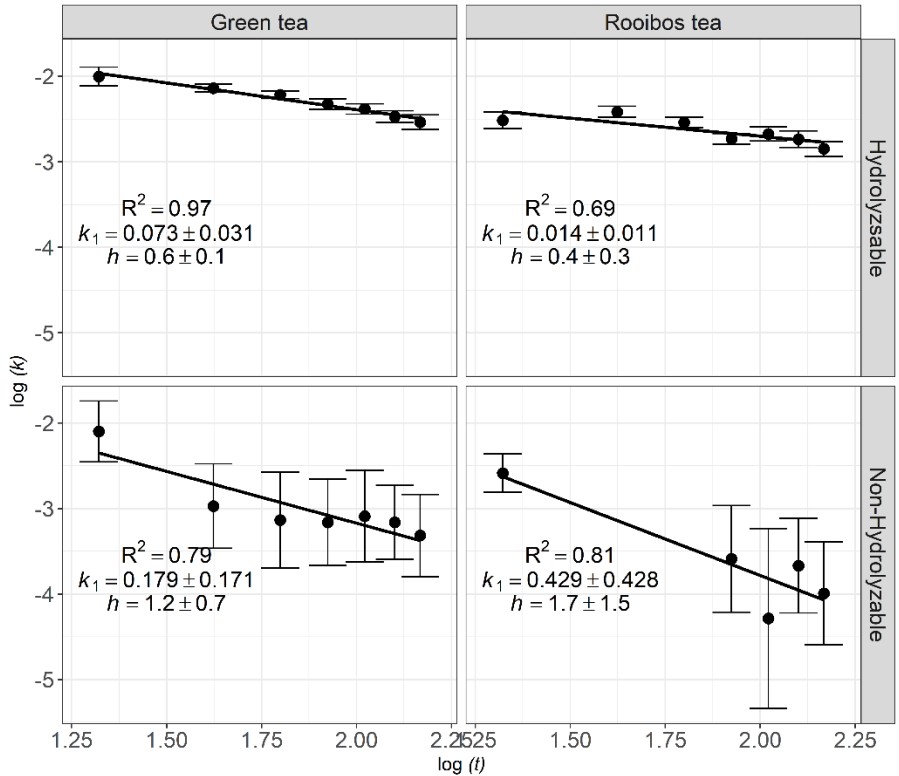

**Figure 8.** Fractal kinetics of the tea's biochemicals compound under field conditions with a 95% compatibility interval.

## 4. Discussion

### 4.1. Effect of Nitrogen Fertilization Regime on Organic Matter Decomposition

The nitrogen fertilization regime did not affect TBI decomposition rates in line with several studies showing little or no effect of nitrogen additions on the decomposition rate of organic matter in agroecosystems [26,68–70]. Increasing the nitrogen dosage may inhibit carbon turnover and decomposition of organic matter under annual cropping [71], especially during the last stage of decomposition [72]. Riggs and Hobbie [73] reported that nitrogen additions decreased the decomposition rate by 31% and microbial respiration by 21% in grassland soils. Laskowski and Berg [74] found an important impact of nitrogen on lignin decay: the higher the nitrogen dosage, the stronger the inhibition of ligninase.

On the other hand, the nitrogen regime impacts the vegetative growth of cranberry plants, possibly leading to overgrowth favorable to carbon accumulation at the expense of berry yield and quality [32,75]. Cranberry sandy soils are acidic and fertilized with acidifying ammonium sulfate in conventional farming systems [76], contributing to restrained microbial activity and carbon decomposition rates [12,70]. Indeed, Min et al. [77] found that ammonium-based fertilizers reduced $CO_2$ fluxes in acid soils (pH = 5.7). Microbial activities are low in strongly acidic [78] and low-nutrient soils [79]. Bacteria are less active in acid soils near pH 6.0, while fungi are more tolerant to low pH, but generally less active at soil pH below 5.0 [76].

### 4.2. Tea Bag Index

The tested cranberry agroecosystems were in the low range of decomposition rates and in the high range of organic matter stabilization factors among terrestrial ecosystems [42]. Mass loss was 59% for green tea and 26% for rooibos tea after 90 days of incubation compared to 63% and 22%, over one year, respectively, in the slow-C decomposing restored peatland ecosystems studied by Macdonald et al. [60]. Mass loss of cranberry litter was 5% of initial mass over 90 days, which is much less than the 51% averaged over one year by Macdonald et al. [60] across four peatland grass species.

The hydrolyzable fractions of green tea (90.5 ± 2.4%) and rooibos tea (82.9 ± 1.7%) using the van Soest method [58] differed from those reported by Keuskamp et al. [44] (84.2% and 55.2%, respectively), who used the Ryan et al. [80] method (Table 3). Nevertheless, the biochemical composition was still closely related to litter decomposition rates. Cranberry litter contained by far the largest proportion of recalcitrant lignin-cutin materials and decomposed at a rate that was an order of magnitude lower compared to the tea litters (Table 3).

The litter C:N ratio is often thought to control litter decomposition in acid soils of conifer forest ecosystems [81] and other biomes [35]. Despite similar C:N ratios in the first experiment, the carbon decomposition rate of the cranberry litter was much lower compared to that of the rooibos tea due to differential van Soest biochemical compositions and differences in particle size distribution that controlled the contact between the substrate and catalytic agents.

### 4.3. Fractal Kinetics

Experimental studies on reaction kinetics may provide evidence for anomalous patterns showing time-dependent reaction rates [48]. Fractal coefficients of green and rooibos teas (Figure 7) where within the 0–1 range suggested by Kopelman [48]. We found fractal coefficients ranging from 0.5 to 0.7 for green tea and 0.0 to 0.4 for rooibos tea in results presented by Keuskamp et al. [44] and Duddigan et al. [62]. First-order kinetics also differed markedly between labile and recalcitrant carbon pools as reported elsewhere [82–85].

Soil and organic matter are fractal objects [86,87] of varying particle size distribution and biochemical composition [85,88]. Carbon accessibility to microbial attacks may change through time due to the initial "priming effect" of labile carbon compounds [89–91], and to the buildup of biological, chemical, and physical protection mechanisms developing in the soil through time [10]. While Keuskamp et al. [44] assumed that the stabilization

factor was regulated by environmental conditions, fractal coefficients also provided additional evidence that the contact between the litters and their immediate environment regulated the tea decomposition rates. After 90 days incubation time, green tea was still not fully decomposed, possibly leading to biases in the estimation of the stabilization parameter. Fractal kinetics could make an additional parameter to account for the specific environmental conditions in slow-decomposing ecosystems.

Indeed, fractal patterns have allowed accounting for the non-linear decreasing rate of soil organic carbon decomposition through time in agricultural [2] and forest ecosystems [47]. At the time scale of TBI monitoring, the green and rooibos teas behave like fractal objects, limiting access to microbial and enzymatic attacks within a single growing season. The green tea particles tended to agglomerate and pack through time in tea bags. The contact between the organic material and its immediate environment was likely facilitated by non-polar extractable compounds [44], particle enmeshing, and microbial products synthesized during litter decomposition [92].

If fractal-like kinetics occur, then the decomposition rate $k_1$ is maximum at the onset of the incubation where $h = 0$ (hence, $t^{-h} = 1$). Indeed, if $h \to 0$, then the reaction rate gets closer to the classical first-order kinetics [48]. If $0 \leq h \leq 1$, then the reaction rate decreases with time. The decomposition rate decreases non-linearly with time as regulated by the fractal coefficient, which depends on tissue structure, biochemical composition, and protection mechanisms.

The fractal coefficient of the non-hydrolyzable fraction was $h \approx 1$, indicating considerable deceleration of the decomposition rate through time for that fraction. Lignocellulose is a complex compact material made of chemically bound cellulose, lignin, and hemicellulose [93]. It is recalcitrant to decomposition, requiring the synergistic action of a broad spectrum of microorganisms and enzymes. For the green and rooibos tea non-hydrolyzable fraction, the fractal coefficients were 1.2 and 1.7, respectively (Figure 8), but did not differ significantly from one (95% compatibility interval). Few studies reported $h \geq 1$ [50].

## 5. Conclusions

The nitrogen fertilization regime did not impact TBI, allowing us to derive TBI values specific to cranberry stands, whatever the nitrogen fertilization regime. The TBI of cranberry agroecosystems was in the low range among terrestrial ecosystems. Cranberry litter decomposed at much lower rate than rooibos tea due to higher content of recalcitrant biochemical fractions, larger particle size, and compact tissue structure. Fractal-like first-order kinetics was found to control the decomposition rate of green and rooibos teas under the environmental conditions of the cranberry production. There is a high potential for carbon accumulation in cranberry agroecosystems following the regular burial of cranberry litter recalcitrant to decomposition and to acidic soil conditions, confirming field observations on litter accumulation between sand layers.

**Author Contributions:** Conceived and designed the experiment: W.D.-Y., S.-É.P. and L.-É.P.; performed the experiment: W.D.-Y. and É.P.; analyzed the data: W.D.-Y. and S.-É.P.; wrote the first draft of the manuscript: W.D.-Y.; supervisors: S.-É.P. and N.Z.; All authors have read and agreed to the published version of the manuscript.

**Funding:** This collaborative research project was supported by Les Atocas de l'Érable Inc., Les Atocas Blandford Inc., La Cannebergière Inc., the Natural Sciences and Engineering Research Council of Canada (RDCPJ-469358-14) and Agriculture Agri-food Canada (AAFC-1555).

**Institutional Review Board Statement:** Not applicable.

**Informed Consent Statement:** Not applicable.

**Data Availability Statement:** Research Data Available at https://bit.ly/3fUZySk.

**Conflicts of Interest:** The authors declare that they have no known competing financial interests or personal relationships that could have appeared to influence the work reported in this paper.

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
