# Peer review of "Tea Bag Index to Assess Carbon Decomposition Rate in Cranberry Agroecosystems"

_soilsystems, doi:10.3390/soilsystems5030044_

Round 1

Reviewer 1 Report

Interesting article. It deals with the issue of the rate of soil organic matter mineralization in ecosystems, which is inherently associated with the sequestration of organic carbon and the possible reduction of the greenhouse effect. The use of TBI is a universal, relatively simple and worth disseminating method that enables the comparison of litter decomposition rates for various habitat conditions. The authors noted that the computation of TBI stabization factor may be potentially biased because green tea was not fully decomposed and fractal kinetics could be used as additional index to compare agroecosystems. Moreover, the study also analyzed the effect of nitrogen fertilization on the rate of litter decomposition. The work is well-edited, minor corrections are recommended (detailed notes are given in the form of comments in text), suitable for printing.

Author Response

The following paper submitted to Soil Systems has been corrected to address reviewers’ comments:

Tea Bag Index to assess carbon decomposition rate in cranberry agroecosystems

Dossou-Yovo, W., Parent, S.-É., Ziadi, N., Parent, E., and Parent, L.E.

Question 1: Table 1: Comment on superscript.

Response1: It is now corrected in the manuscript (Mehlich-3 extractable mg element kg-1)

Question 2: Table 2: Comment on particle size fraction

Response 2: It is now corrected in the manuscript as:

Tea type

Size fraction (mm)

Particle size distribution (%)

Green tea

>2

32.1

1-2

26.2

0.5-1

5.9

0.25-0.50

1.7

<0.25

34.1

Question 3: Comment on line 190

Response 3: You’re right. I have missed the log. I correct it in the manuscript:

log(kt) = -h log(t) + log(k1)

Question 4: Comment on table3, line 224

Response 4: I have added year “performed in 2017” to the table3 title.

Question 5: Comment on line 251

Response 5: I have checked my data computation; everything is now fine about figure 6. There is no significant difference between decomposition rates between Table 3 and figure 6 after 90 days of incubation.

Question 6: Comment on line 257

Response 6: The fractal coefficient of green tea 0.6 not 0.7. It was my mistake. It has been corrected in the manuscript

Question 7: Comment on line 271

Response 7: You are right. I removed “cranberry litter.” I also added the experimentation performing date as you asked. “performed in 2018”

Question 8: Comment on line 301

Response 8: I have corrected the hydrolysable fraction value of Keuskamp et 2013. The values are 84.2% and 55.2% for green tea and rooibos tea, respectively.

Question 9: Comment on line 335

Response 9: Where h = 0 instead of t = 1. I am sorry for this. Thank you for the remark.

Reviewer 2 Report

In the current manuscript, the author studied the effect of nitrogen addition of plant litter decomposition (two type of tea and cranberry residues), and compared the decomposition process of tea in cranberry ecosystem with other ecosystems. The results showed that the decomposition of cranberry is lower compared with two teas, and the tea decomposed much slower in the study site than other ecosystems. This topic is interesting, and the English is perfect. Below are line by line comments.

Ln13: based on the data obtained in the experiment, the effect of nitrogen addition on plant litter decomposition is insignificant. But I think you should not propose the hypothesis based on the results. Generally, nitrogen addition can decrease plant litter decomposition through inhibiting microbial activity or through nutrient mining theory. Thus, we can expect a lower decomposition rate after nitrogen addition before the experiment was conducted.

Ln20: can you explain why the plant litter decomposition rate was not affected by nitrogen addition? Maybe the added nitrogen is very easy to leach due to the cover of sand on the soil?

Ln28: potentally?

Ln63: add a period befor “The green tea”.

Ln67-68: as far as I know, the first-order kinetics model does not mean that the reaction rate is constant over time. Instead, the reaction rate is determined by the decomposition rate constant and the concentration of substrate. dx/dt=-kx, then, wen can get the commonly used Xt=X0*exp(-k*t), where k is the decomposition rate constant. You need to know the difference between decomposition rate and decomposition rate constant. If the data in fig. 6 means decomposition rate, but not the decomposition rate constant, we can expect a negative relationship between decomposition rate and incubation time in the field, since the substrate concentration declined with time.

Ln76-77: why did you conduct the nitrogen addition treatment if you hypothesized that nitrogen addition did not change the decomposition of plant litters?

Ln120: “Mehlich-3 extractable mg element kg-1”, “-1” should be superscript.

Ln141: a sizable study, we can image that the data obtained in this experiment is very reliable.

Ln150-151: the description about nitrogen addition treatment is not clear. Did you conduct all the nitrogen addition treatment in the four study sites? What is the nitrogen form? How and when the nitrogen addition was applied? Once each year? In addition, the decomposition rate of plant litter under different nitrogen addition levels were not shown in the manuscript. We need to know those information, even the nitrogen effect was not significant.

Ln158: I am puzzled by the initial weight of the plant litters. Did you record all the weight in each litter bags? How did you mark those litter bags? Or 1.7-1.9 g is just the range of the plant litters, and you used an average value as the initial weight of plant litters (for example 1.8 g)? If so, no wonder that there is a large uncertainty in the decomposition rate of cranberry residue. This value is very important in the plant litter decomposition experiment, thus, it should be weighted very carefully. Generally, I weight the initial litters in the range of 1.998-2.002 g, and then use 2.0 g as the initial plant litter weight.

Ln276-284: can you explain more about why the nitrogen effect on plant litter decomposition is not significant in your study site. Is it because of the poor retention ability of nitrogen due to sand application? Additionally, can you provide some information of soil inorganic nitrogen concentration under different nitrogen addition levels?

Ln298-299: why? What is the reason for the inconsistence of mass loss between your study and previous results?

Sum up, this manuscript might be accepted after careful revision.

Author Response

The following paper submitted to Soil Systems has been corrected to address reviewers’ comments:

Tea Bag Index to assess carbon decomposition rate in cranberry agroecosystems

Dossou-Yovo, W., Parent, S.-É., Ziadi, N., Parent, E., and Parent, L.E.

Question 1: Ln13: based on the data obtained in the experiment, the effect of nitrogen addition on plant litter decomposition is insignificant. But I think you should not propose the hypothesis based on the results. Generally, nitrogen addition can decrease plant litter decomposition through inhibiting microbial activity or through nutrient mining theory. Thus, we can expect a lower decomposition rate after nitrogen addition before the experiment was conducted.

Response 1: “While biomass production depends on N fertilization, almost intact twigs and leaves remain in subsurface layers, indicating carbon accumulation and slow decomposition rate whatever the nitrogen fertilization regime.”

Changed to: However, almost intact twigs and leaves remain in subsurface layers, indicating slow litter decomposition rate.

Question 2: Ln20: can you explain why the plant litter decomposition rate was not affected by nitrogen addition? Maybe the added nitrogen is very easy to leach due to the cover of sand on the soil?

Response 2: We added “possibly due to N leaching or strong acidic soil condition”

Question 3: Ln28: potentially?

Response 3: “potentially” deleted

Question 4: Ln63: add a period before “The green tea”.

Response 4: We added “after 90 days of incubation”

Question 5: Ln67-68: as far as I know, the first-order kinetics model does not mean that the reaction rate is constant over time. Instead, the reaction rate is determined by the decomposition rate constant and the concentration of substrate. dx/dt=-kx, then, wen can get the commonly used Xt=X0*exp(-k*t), where k is the decomposition rate constant. You need to know the difference between decomposition rate and decomposition rate constant. If the data in fig. 6 means decomposition rate, but not the decomposition rate constant, we can expect a negative relationship between decomposition rate and incubation time in the field, since the substrate concentration declined with time.

Response 5: “Decomposition rate” was changed by “Instantaneous decomposition rate constant” in Figure 6. As reported by Kopelman (1988), “Classical reaction kinetics has been found to be unsatisfactory when the reactants are spatially constrained on the microscopic level by either walls, phase boundaries, or force fields. Recently discovered theories of heterogeneous reaction kinetics have dramatic consequences, such as fractal orders for elementary reactions, self-ordering and self-unmixing of reactants, and rate coefficients with temporal memories." In natural heterogeneous systems, there is no means to mix the substrate and the catalyst (enzymes), limiting the contact between them. Such behavior is described by fractal kinetics.

  1. Kopelman, Fractal reaction kinetics, Science. 241 (1988) 1620-1626.

Question 6: Ln76-77: why did you conduct the nitrogen addition treatment if you hypothesized that nitrogen addition did not change the decomposition of plant litters?

Response 6: We added in introduction (l. 44-45):

Nitrogen applied at rates that may vary in the range of 15 to 60 kg N ha-1 is the nutrient impacting most vegetative growth, sometimes reaching excessive biomass production [Jamaly et al., 2021].

Reza Jamaly; Serge-Étienne Parent; Léon E. Parent. Fertilization and Soil Nutrients Impact Differentially Cranberry Yield and Quality in Eastern Canada Horticulturae 2021, Volume 7, Issue 7, 191

Question 7: Ln120: “Mehlich-3 extractable mg element kg-1”, “-1” should be superscript.

Response 7: Done.

Ln141: a sizable study, we can image that the data obtained in this experiment is very reliable.

Thank you. Big work indeed.

Question 8: Ln150-151: the description about nitrogen addition treatment is not clear. Did you conduct all the nitrogen addition treatment in the four study sites? What is the nitrogen form? How and when the nitrogen addition was applied? Once each year? In addition, the decomposition rate of plant litter under different nitrogen addition levels were not shown in the manuscript. We need to know such information, even the nitrogen effect was not significant.

Response 8: We added (ln 153-158): “Nitrogen sources were ammonium sulfate in the conventional sites or fish emulsions (6-1-1) in the organic site. Nitrogen was applied at four occasions during the season: 15% at early flowering (29 June to 2 July), 35% at 50% flowering (8 to 11 July), 35% at 50% fruit set (16 to 19 July), and 15% 1 to 2 weeks after the third application. Other nutrients were applied according to local fertilization trials or regional fertilizer recommendations.”

Question 9: Ln158: I am puzzled by the initial weight of the plant litters. Did you record all the weight in each litter bags? How did you mark those litter bags? Or 1.7-1.9 g is just the range of the plant litters, and you used an average value as the initial weight of plant litters (for example 1.8 g)? If so, no wonder that there is a large uncertainty in the decomposition rate of cranberry residue. This value is very important in the plant litter decomposition experiment, thus, it should be weighted very carefully. Generally, I weight the initial litters in the range of 1.998-2.002 g, and then use 2.0 g as the initial plant litter weight.

Response 9: Ln. 165: we added “Each bag and litter were weighed and tagged.”

Question 10: Ln276-284: can you explain more about why the nitrogen effect on plant litter decomposition is not significant in your study site. Is it because of the poor retention ability of nitrogen due to sand application? Additionally, can you provide some information of soil inorganic nitrogen concentration under different nitrogen addition levels?

Response 10: Some literature reports the decreasing rate of nitrogen addition on organic matter decomposition. In cranberry ecosystems, nitrogen is the primary fertilizer used by growers. But sand quality, burying of organic matter by sanding, and strong acidic soil conditions may inhibit the effect of nitrogen fertilization.

Unfortunately, we did not quantify the soil inorganic nitrogen concentration. We only provided the pressure of inorganic nitrogen fertilization.

Question 11: Ln298-299: why? What is the reason for the inconsistence of mass loss between your study and previous results?

Response 11: Originally cranberry bog is a peat-based, but for industrial production by growers, cranberry bogs differ from original bogs because the upper layers of natural vegetation have been removed, the soil has been modified by the addition of a sand layer, and cranberry plants have been introduced. The difference can be explained by the fact that they are different ecosystems that can compared as Keuskamp did across terrestrial ecosystems.

Sum up, this manuscript might be accepted after careful revision.

Thank you.

This manuscript is a resubmission of an earlier submission. The following is a list of the peer review reports and author responses from that submission.

Round 1

Reviewer 1 Report

General Remarks

Dear authors,

Your research dealing with the decomposition of organic matter from cranberry production sites using the TBI, is interesting, especially with respect to carbon storage in soils, probably as well in context of climate change (you did not mention this). Your results should be made available for the community! 

I for me learned quite a lot, starting with the specially designed soils for cranberry production!

However, I found several points that in my opinion need to be addressed before publication. This starts with the title. I am really not sure if “using fractal kinetics” is correct, I would think, you tested the organic matter decomposition for fractal kinetics (as found for green and rooibos tea). Or did I misunderstood something here (s. Special Remarks)? Another point was the experimental design that not was exactly the same in the two years (s. Special Remarks). Is there a reason for these differences? Finally, as a remark to the use of English language, I would think some more articles should be added, I made some suggestions here (s. Special Remarks).

Overall, I would think at least some minor revision should be made before publication.

Special Remarks

For my Special Remarks, please see the attached pdf document I added my comments to!

Author Response

Most the responses to reviewers’ comments are in the Ms using “Track Changes”.

Response to reviewer 1 comments

Comment on line 2: The manuscript reports on fractal kinetics in 2.4.3, 3.2 and 4.3

Comment on line 22: We detected fractal patterns in Keuskamp et al. 2013 and Duddigan et al.2020.  The decreasing rate of organic matter decomposition through time indicates that surfaces became less accessible to microbial attack through time. Such decreasing rate is ddescribed by fractal coefficient h (0 ≤ h ≤ 1, t ≥ 1) as suggested by Kopelman, 1988. The higher is fractal coefficient, the greater are the protective measures against organic matter decomposition.

Comment on line 45: I do not think any word is missing. I am talking about carbon sink as carbon reserve.

Comment on line 122: Tea bags buried in cranberry stands have serious issues with unfound or bags damaged during unearthing. This is why we used enough bags (3792) in the second experiment to ensure that we have enough samples for our analyses.

Comment on line 243: Yes, I mean ammonium sulfate. Urea can be converted readily into ammonium in the soil.

Comment on line 312: Here, we compare materials with similar C/N ratio. The C/N ratio of green tea (13) is much lower than that of rooibos (53) and cranberry litter (55)

Reviewer 2 Report

Is this paper assessing decomposition rates in cranberry stands or how the TBI works in cranberry stands? the title and abstract are contradictory in answering the above question.

Intro too short and should cover more about decomposition and the agricultural practice of growing cranberries. please discuss what you mean by first order kinetics (unclear if this even belongs in this paper

Methods are lacking and needs much work before publication. TBI is not explained clearly. How did you seal the bags? Did you use bags only as a control to see if bags lost or gained weight (Sediment can work its way into the bags).

Improve on intro and methods and the results and discussion may be ok. However, the first-order kenetics seems really out of place here and should be better explained if it is to remain in this manuscript.

Author Response

Response to reviewer 2 comments

Point 1: Is this paper assessing decomposition rates in cranberry stands or how the TBI works in cranberry stands? the title and abstract are contradictory in answering the above question.

Response 1: This paper assesses decomposition rate of litters in cranberry stands using TBI that allows comparing agroecosystems worldwide according to decomposition rate and stabilisation factor by using two standardized litter types (green tea and rooibos tea).

Point 2: Intro too short and should cover more about decomposition and the agricultural practice of growing cranberries. please discuss what you mean by first order kinetics (unclear if this even belongs in this paper)

Response 2:

  • I added more content in introduction as suggested.
  • We used TBI to compare carbon decomposition rate in cranberry agroecosystem to that of other terrestrial agroecosystems documented by Keuskamp et al. (2013). Litters were standardised tea materials (green and rooibos tea) and cranberry residues. Organic matter decomposition rate is generally described by first order kinetics. As expected, green tea decomposed more rapidly than rooibos and cranberry litter. Despite similar C/N ratio for cranberry residues (55) and rooibos tea (53), cranberry litter decomposed at much lower rate than rooibos due to particle size distribution, compact tissues structure and recalcitrant biochemical components.
  • Fractal kinetics (Kopelman, 1988) slows down first order kinetics.

      First order kinetic:  ln (Mt/M0) = - kt, k assumed to be constant.

Fractal kinetic: kt = k1 t – h, decreasing with time, hence ln (Mt/M0) = - k1 t 1- h; where h is fractal coefficient

  •  

Point 3: Methods are lacking and needs much work before publication. TBI is not explained clearly. How did you seal the bags? Did you use bags only as a control to see if bags lost or gained weight (Sediment can work its way into the bags).

Response 3: Tea bags were sealed with a bag sealer. We used bags as control and others were incubated in the field. More details in the methods section

Point 4: Improve on intro and methods and the results and discussion may be ok. However, the first-order kenetics seems really out of place here and should be better explained if it is to remain in this manuscript.

Response 4: first order kinetics is addressed above.

Reviewer 3 Report

The authors of “Organic matter decomposition in cranberry stands using fractal kinetics and Tea Bag Index” present a study on the decomposition of green tea, rooibos tea as compared to cranberry litter in a cranberry production system. They use the tea bag index on the one hand to evaluate the influence of nitrogen dosing on OM degradation and in a second trial perform degradation kinetics on model OM in tea-bags.

While in principal the merit of the study could be to demonstrate and explain mechanisms of carbon storage in cranberry production systems its is currently not well enough written in order to follow the methods applied. Also in the introduction section state-of-the art concerning fractal-like kinetics can be improved and cleary describe as compared to first oder kinetic with various pools.

All equations and data treatments must be given and presented more clearly and all parameters clearly defined in the methods section. Number of replication and standard deviations for each site and sampling strategies for each site has to be presented and is currently missing.

Some figures and tables need to be improved concerning quality and description in figure cations. Raw data need to be made accessible.

Discussion need to refer to own data, figures and tables – which is currently not stringently the case.

The conclusions should refer back to the initial hypothesis and answer those. When the main conclusion is that cranberry fields can contribute to C-sequestration then a model/Back-of the envelope calculation based on own data and giving all assumptions should be performed in the discussion section that calculates a rate of C storage potential per area and year.

At the moment, this paper needs considerable work, first and foremost on a more complete and comprehensive way to present data and make transparent which and how many data are used with which data treatment to represent which time-point or data-treatment. A special emphasis should lie on presenting the study design clearly: how were tea bags distributed in the fields, how many were recovered, are data derived representative of the fields and so forth (see many point-by-point comments)?

Given the authors were willing to completely revise the material and data presentation, this could be a nice and interesting case study for OM storage potential in cranberry production systems.

L 77-79 additional soil names according to WRB classification would be very nice!

Fig. 1 : Image quality is poor and needs to be enhanced. Left text is not readable. Very pixely.

L90 What was your sampling strategy? How did you make sure that your single data points described in Table 2 represent the respective sites?

L92: Why not measure pH in water? I think the calculated value can only serve as approximation and needs to be presented as such. “appricimative values for pH in H2O are obtained by using….”

L99: Please be specific in describing which parameters were detected with the method described.

Table 1:

-texture data sum up to between 933 and 1001 g kg-1. How can this variation be explained? Error/loss should be indicated somewhere?

-instead of repeating “sand” so often I would suggest you write “sand” vertically i.e tilt one  90°counterclockwise across all fractions

- what is the difference in significance/meaning of S by CNS-Analyzer and Mehlich-3 – indicate as plant available, total in method section above (see comment L99)….?

(move to results section and comment on results)

L108: How did you determine C:N ratio here?

Table 2 and L 114: How do the starting tea bag C:N and distribution values compare to literature data? (move to results section and comment on results)

L121 How was the distribution of tea bags between field? Pattern? How many per site? Covering which area? Were all buried and sampled at the same time?

L122 Do not understand: Why was it important that the number was much larger than 110 and 234 (exact numbers)?

What is the relationship between soil analysis and tea bag analysis?

L126-133 What is the experimental design of various N fertilization in relation to the plots? All fertilization rates on all plots? Maybe good to provide a schematic overview of plots in relation to N treatents , tea bags buried and soil samples taken.

(I would present the general design and especially the tested hypothesis at the end of introductory sectionà add after l 66 as an objective)

Ll135 ff. I suggest you present this info on study design before sampling details. Now you said 432 bags across all sites but above it were >3000. 8cm depthà redundancy

Table 3 M 90 days needs to be defined with units , ln (M90/M0)/90 correct? First-order rate kontatant here? Indicate in table capture. Where is the fractals?

L168: Which model was evaluated with linear mixed-effects?

L186 remove error:”(Error! Refrence source not found)”

L188 “stabilization factor” – where do I find them and how is this factor defined; equal to TBI?

Figure 3 and 4: How do you need an lme of you want to test a relationship between only TBI and N dosage – keep it simple! Linear regression with significance testing and everyone can understand what you have been doing. Or otherwise, if more complex, please explain in more detail for the outsider and  present raw data! Also indicate somewhere, what the coefficients means/how you interpret them.

Table 3: You had so many teabags but you present data without standard deviation. I need a table that also gives n, the number of samples analyzed and the standard deviation or if this should be the data from initial single measurements then I would need this info in the Table caption

L199 S and k are mentioned out of the blue here. Where are tables with raw data, factors per sample? How is the stabilization factor defined an derived? If S= TBI then you need to defined this above more clearly. If you only write one sentence to comment on Figure 5 maybe you can delete this. If there were more ecological remarks on this Figure I would be interested to here them and learn about the significance and possibly of using the TBI to interpret soil ecological processes!?

Continue L207

L207 ff. Define definition and decomposition rate calculation properly (reaction rate = decomposition rate? eq. 2- not clearly written) in methods section (at the moment hidden in Table 3 (would remove from table 3 and integrate in methods text- present equation)

What does two-point decomposition rates as compared to fractal decomposition rate mean?

Figure 6: Based on first oder kinetic (eq. 2)? Please specify in figure caption. Have the C:N ratios been determined at each timepoint? Could you plot decomposition rates vs. C:N ratios in for all litter types. Does cranberry C:N remain at C:N 50?

L224: move to methods section along with corresponding equations.

Figure 7 and 8: indicate in Figure caption meaning of  log (k) and log (t), ka and h. Also, it should be clear already from the meothods section which is what – which at the moment is not the case. Indicate also the nature of the (linear regression?) line plotted along with single (?) data points? Are these really singe data or means This should be clear from methods section! Corresponding raw data should be made accessible to reader/reviewer.

L234ff. in Discussion section: A general introductory paraprah L237-242 should be moved to introduction or after mentioning of own findings in oder to “discuss” them.. Relaltion of discussion to own data should be clear.

The whole section does not relate at all to own data. But what do the literature finding mean for your own data. Have they been confirmed? Are they different? In which way and why were they different? If you do not comment on effects of nitrogen fertilization that you determined, then move this section to intro as background.

In the whole section you should refer to your data also by referencing to figures and tables, which is not done once at the moment!

L267 correct “lesser” to “less” oder “lower”

L269 This is interesting and what were C:N ratios or in which figure or table can I find them?

L273-286 This is intro! No own data!

L 284 What is a? define in method or in intro and refer here to corresponding equation

L287 refer to presentation of these data! Also apply to the rest of the discussion

L294 (refer this statement to own data, observation or literature)

L306 Non-hydrolysable fractions of all litter types? Please be more specific to which data you refer

L312 How did the C:N ratios change over time? Rooibos decreased probably and canbrery litter remained the same due to high content in lignocellulose – where are the data to show this clearly?

Particle sizes before and after?

L318 What is the role of peatland in your study – this is totally not clear having arrived here and should be clearly presented in results and discussion

Refer back to your hypothesis in the conclusion section and answer them clearly and concisely

Author Response

Most the responses to reviewers’ comments are in the Ms using “Track Changes”.

Response to reviewers 3 comments

The authors of “Organic matter decomposition in cranberry stands using fractal kinetics and Tea Bag Index” present a study on the decomposition of green tea, rooibos tea as compared to cranberry litter in a cranberry production system. They use the tea bag index on the one hand to evaluate the influence of nitrogen dosing on OM degradation and in a second trial perform degradation kinetics on model OM in tea-bags. 

Point 1: While in principal the merit of the study could be to demonstrate and explain mechanisms of carbon storage in cranberry production systems its is currently not well enough written in order to follow the methods applied. Also in the introduction section state-of-the art concerning fractal-like kinetics can be improved and clearly describe as compared to first order kinetic with various pools.

Response 1: I explained more the purpose of fractal kinetics in introduction.

Point 2: All equations and data treatments must be given and presented more clearly and all parameters clearly defined in the methods section. Number of replication and standard deviations for each site and sampling strategies for each site has to be presented and is currently missing.

Response 2: The TBI formulae were referred to the Keuskamp article. As suggested, I added equations for TBI, first order kinetics and fractal kinetics.

The number of replications and SD has been presented in the methods as well as sampling strategies.

Point 3: Some figures and tables need to be improved concerning quality and description in figure cations. Raw data need to be made accessible.

Response 3: I improved the quality of figure 6. In supplementary materials I provided a link to access to all data and statistical analysis in the R environment.

Point 4: Discussion need to refer to own data, figures and tables – which is currently not stringently the case.

Response 4: I added more reference to figures and tables into the discussion section.

Point 5: The conclusions should refer back to the initial hypothesis and answer those. When the main conclusion is that cranberry fields can contribute to C-sequestration then a model/Back-of the envelope calculation based on own data and giving all assumptions should be performed in the discussion section that calculates a rate of C storage potential per area and year.

Response 5: Several parameters and factors contribute to C storage: decomposition rate, stabilisation factor, residencee time, grain-size distribution, microbial activity, cultural practice and burial rate. The stabilisation factor is an indicator of C storage potential (see screen shot and figures in Keuskamp et al. 2013 below).

Screen shot from Keuskamp et al. 2013

Point 6: At the moment, this paper needs considerable work, first and foremost on a more complete and comprehensive way to present data and make transparent which and how many data are used with which data treatment to represent which time-point or data-treatment. A special emphasis should lie on presenting the study design clearly: how were tea bags distributed in the fields, how many were recovered, are data derived representative of the fields and so forth (see many point-by-point comments)?

Response 6: see section 2.4 of the manuscript.

 Point 7: Given the authors were willing to completely revise the material and data presentation, this could be a nice and interesting case study for OM storage potential in cranberry production systems.

Response 7: The method section has been revised as you suggested. Any additional information is available in the link provided in supplementary material.

Point 8: L 77-79 additional soil names according to WRB classification would be very nice!

Response 8: the USDA classification is recognized worldwide (l. 77-79). I am more familiar with that soil classification system.

Point 9: Fig. 1 : Image quality is poor and needs to be enhanced. Left text is not readable. Very pixely.

Response 9: I improved image quality.

Point 10: L90 What was your sampling strategy? How did you make sure that your single data points described in Table 2 represent the respective sites?

Response 10: Table 2 describes the initial (control) the particle size of the litters (rooibos, green tea and cranberry residues)

Point 11: L92: Why not measure pH in water? I think the calculated value can only serve as approximation and needs to be presented as such. “appricimative values for pH in H2O are obtained by using….”

Response 11: I reported measured pHCacl2.

Point 12: L99: Please be specific in describing which parameters were detected with the method described.

Response 12: Ok, I added more content in the method.

Point 13: Table 1:

-texture data sum up to between 933 and 1001 g kg-1. How can this variation be explained? Error/loss should be indicated somewhere?

Response 13: Soil texture was measured by two methods conducted sequentially: hand sieving and sedimentation. This variation can be explained by error or loss. I standardized the results as per 1000 g kg-1.

Point 14: -instead of repeating “sand” so often I would suggest you write “sand” vertically i.e tilt one  90°counterclockwise across all fractions

Response 14:  done

Point 15: - what is the difference in significance/meaning of S by CNS-Analyzer and Mehlich-3 – indicate as plant available, total in method section above (see comment L99)….?

(move to results section and comment on results)

Response 15: S by CNS-Analyser is total soil Sulfur content; S by Mehlich-3 is plant available sulfur content.

Point 16: L108: How did you determine C:N ratio here?

Response 16: We used Leco CNS analyser to quantify C and N and computed C/N ratio

Point 17: Table 2 and L 114: How do the starting tea bag C:N and distribution values compare to literature data? (move to results section and comment on results)

Response 17:  According to Keuskamp et al. (2013), C/N ratio of green tea is 12.229±0.129 and that of rooibos tea is 52.870±1.841. For cranberry litter, the C/N ratio (55) was lower than that ‘Stevens’ uprights (60-70) measured by vanden Heuveln and Davenport (2006) at midfruit development.

Point 18 L121 How was the distribution of tea bags between field? Pattern? How many per site? Covering which area? Were all buried and sampled at the same time?

Response 18: see M&M section

Point 19: L122 Do not understand: Why was it important that the number was much larger than 110 and 234 (exact numbers)?

Response 19: tea bags are fragile and have serious issues in the tough and dense cranberry stands due to unfound or damaged bags during unearthing. 

Point 20: What is the relationship between soil analysis and tea bag analysis?

Response 20: Soil analysis was about cranberry soil characteristics while litter analysis is about our treatment. Soil characteristics allows to explain low TBI decomposition and high stabilisation factor.

Point 21: L126-133 What is the experimental design of various N fertilization in relation to the plots? All fertilization rates on all plots? Maybe good to provide a schematic overview of plots in relation to N treatments, tea bags buried, and soil samples taken.

Response 21: more details in the M&M section. According to vanden Heuveln and Davenport (2006), the N fertilization influence the C/N ratio, hence potentially decomposition rate of organic matter.

Point 22: (I would present the general design and especially the tested hypothesis at the end of introductory sectionà add after l 66 as an objective)

Response 22: I have added your request in the introduction session

Point 23: Ll135 ff. I suggest you present this info on study design before sampling details. Now you said 432 bags across all sites but above it were >3000. 8cm depth redundancy

Response 23: 3792 bags were the total of the two years of experimentation: 432 in year 1 and 3360 bags in year 2.

Point 24: Table 3 M 90 days needs to be defined with units, ln (M90/M0)/90 correct? First-order rate constant here? Indicate in table capture. Where is the fractals?

Response 24: I added the unit “Days” in table 3. The fractal kinetics does not concern two points (0-90 days) kinetics. I added more explanation on the fractal kinetics in introduction, material and methods, results and discussion sections.

Point 25: L168: Which model was evaluated with linear mixed-effects?

Response 25: TBI parameters (k and S) were analyzed statistically.

Point 26: L186 remove error:”(Error! Refrence source not found)”

Response 26: Done.

Point 27: L188 “stabilization factor” – where do I find them and how is this factor defined; equal to TBI?

Response 27: explained in method section.

Point 28: Figure 3 and 4: How do you need an lme of you want to test a relationship between only TBI and N dosage – keep it simple! Linear regression with significance testing and everyone can understand what you have been doing. Or otherwise, if more complex, please explain in more detail for the outsider and  present raw data! Also indicate somewhere, what the coefficients means/how you interpret them.

Response 28: You are right, but I tried both linear regression and linear mixed model. Because the random effect (variation between sites) was important linear regression was not good enough. With linear mixed model, we can estimate the variation between sites.

Point 29: Table 3: You had so many teabags but you present data without standard deviation. I need a table that also gives n, the number of samples analyzed and the standard deviation or if this should be the data from initial single measurements then I would need this info in the Table caption

Response 29: I provided your request in the table 3. I also provide a link in supplementary material to access all data and statistical analysis in r: https://bit.ly/3fUZySk. I provided standard deviation in the method as you suggested

Point 30: L199 S and k are mentioned out of the blue here. Where are tables with raw data, factors per sample? How is the stabilization factor defined an derived? If S= TBI then you need to defined this above more clearly. If you only write one sentence to comment on Figure 5 maybe you can delete this. If there were more ecological remarks on this Figure I would be interested to here them and learn about the significance and possibly of using the TBI to interpret soil ecological processes!?

Response 30: I provided more detail about TBI computation in the method section.

Point 31: Continue L207

L207 ff. Define definition and decomposition rate calculation properly (reaction rate = decomposition rate? eq. 2- not clearly written) in methods section (at the moment hidden in Table 3 (would remove from table 3 and integrate in methods text- present equation)

Response 31: I have all the equations as requested in the method section

Point 32: What does two-point decomposition rates as compared to fractal decomposition rate mean?

Response 32:

The TBI is an ecosystem-specific index that requires computing results from green tea and rooibos tea. We used two points decomposition rates because we wanted to compare of green and rooibos tea to cranberry litters. That why we used a third bag comprised of cranberry litters.

We found that the decomposition rates of green tea and rooibos tea show fractal kinetics (decreasing rate through time). Also, green tea was not fully decomposed even after 147 days incubation time leading to biases in stabilization index. Fractal kinetic could be a additional index in low litters decompositions agroecosystems.

Point 33: Figure 6: Based on first oder kinetic (eq. 2)? Please specify in figure caption. Have the C:N ratios been determined at each timepoint? Could you plot decomposition rates vs. C:N ratios in for all litter types. Does cranberry C:N remain at C:N 50?

L269 This is interesting and what were C:N ratios or in which figure or table can I find them?

Response 33: No, we did not show C:N ratios at each time point although C and N have been quantified at each timepoint. We were comparing litters decomposition rate according to the initial C:N ratio of tea materials and cranberry litters especially rooibos tea and cranberry litters, which have similar initial C:N ratio.

Point 34: L224: move to methods section along with corresponding equations.

Response 34: Done.

Point 35: Figure 7 and 8: indicate in Figure caption meaning of  log (k) and log (t), ka and h. Also, it should be clear already from the methods section which is what – which at the moment is not the case. Indicate also the nature of the (linear regression?) line plotted along with single (?) data points? Are these really singe data or means This should be clear from methods section! Corresponding raw data should be made accessible to readers

Response 35: fractal kinetic equation kt = k1 t - h        hence, log(kt) = -h log(t) + k1 .

Point 36: L234ff. in Discussion section: A general introductory paraprah L237-242 should be moved to introduction or after mentioning of own findings in oder to “discuss” them.. Relaltion of discussion to own data should be clear.

Response 36: I rearranged that paragraph in discussion after mentioning my own findings.

Point 37: The whole section does not relate at all to own data. But what do the literature finding mean for your own data. Have they been confirmed? Are they different? In which way and why were they different? If you do not comment on effects of nitrogen fertilization that you determined, then move this section to intro as background.

In the whole section you should refer to your data also by referencing to figures and tables, which is not done once at the moment!

L273-286 This is intro! No own data!

Response 37: I rearranged that section by introducing my finding in first place.

Point 38: L267 correct “lesser” to “less” oder “lower”

Response 38: Done

Point 39: L 284 What is a? define in method or in intro and refer here to corresponding equation

Response 39: I added the full term labile (a) and hydrolysable (H) fractions

Point 40: L287 refer to presentation of these data! Also apply to the rest of the discussion

Response 40: Ok I refer it to figure 7

Point 41: L294 (refer this statement to own data, observation or literature)

Response 41: It is an observation, I added in this sentence that it is an observation  

Point 42: L306 Non-hydrolysable fractions of all litter types? Please be more specific to which data you refer

L312 How did the C:N ratios change over time? Rooibos decreased probably and cranberry litter remained the same due to high content in lignocellulose – where are the data to show this clearly?

Response 42: Our data exploration indicates that C:N do not change markedly over time. Our study emphasising that the initial C:N ratio is a limited index to describe litters decomposition while biochemical composition, particle size and compact tissue structure had much impact on litter decomposition especially cranberry litters. I have also provided a link in supplementary material to access all my data and statistical analysis in r: https://bit.ly/3fUZySk

Point 43: Particle sizes before and after?

Response 43: No, just before as control. I make it clear in section 2.3.

Point 44: L318 What is the role of peatland in your study – this is totally not clear having arrived here and should be clearly presented in results and discussion

Response 44: Peatland is an ecosystem documented by Keuskamp et al. (2013). I wanted to present the place of cranberry agroecosystems among ecosystems where TBI was tested.

Point 45: Refer back to your hypothesis in the conclusion section and answer them clearly and concisely

Response 45: I added the hypothesis about the impact of Nitrogen fertilization on TBI in the introduction section to complete the links between hypothesis and conclusions.

Round 2

Reviewer 3 Report

The authors of the manuscript “Organic matter decomposition in cranberry stands using fractal kinetics and Tea Bag Index” revised the title of their presentation to “Tea bag Index to assess carbon release rate in cranberry agroecosystem”.

Several points have been included in the revised version of the manuscript, especially the introduction and some improvement in data presentation, however, in other places, even though stated that a improvement was done (Figure 1) I could not note this – here still poor pixels.

In their answer to the reviewers it would have helped if the authors would have also cited the parts of the manuscript, as they have adapted and pointed out how they changed the manuscript and if they leave it unchanges than say that is wasn’t and why, instead of just explaining e.g. that they like the US classification more (sure it is internationally recognized sytems, so is the WRB – and sometimes it helps a fraction of the readership to have both nomenclatures.

There is still room for improvement, but I now I have to reject the manuscript, as the release rate mentioned in the adapted title appears only in title and introduction and nowhere in this manuscript have carbon release rates been presented or sufficiently clearly explained  to the readers what the relation of TBI and fractal kinetics to the release rates are. The authors should work on their presentation, also again taking into account the comments already given, and present their principally not uninteresting study as a coherent “story”.